# Intestinal FGF15/19 physiologically repress hepatic lipogenesis in the late fed-state by activating SHP and DNMT3A

Young-Chae Kim [1], Sunmi Seok[1], Yang Zhang[2], Jian Ma [2], Bo Kong [3], Grace Guo[3], Byron Kemper[1] & Jongsook Kim Kemper [1✉]

Hepatic lipogenesis is normally tightly regulated but is aberrantly elevated in obesity. Fibroblast Growth Factor-15/19 (mouse FGF15, human FGF19) are bile acid-induced late fed-state gut hormones that decrease hepatic lipid levels by unclear mechanisms. We show that FGF15/19 and FGF15/19-activated Small Heterodimer Partner (SHP/NR0B2) have a role in transcriptional repression of lipogenesis. Comparative genomic analyses reveal that most of the SHP cistrome, including lipogenic genes repressed by FGF19, have overlapping CpG islands. FGF19 treatment or SHP overexpression in mice inhibits lipogenesis in a DNA methyltransferase-3a (DNMT3A)-dependent manner. FGF19-mediated activation of SHP via phosphorylation recruits DNMT3A to lipogenic genes, leading to epigenetic repression via DNA methylation. In non-alcoholic fatty liver disease (NAFLD) patients and obese mice, occupancy of SHP and DNMT3A and DNA methylation at lipogenic genes are low, with elevated gene expression. In conclusion, FGF15/19 represses hepatic lipogenesis by activating SHP and DNMT3A physiologically, which is likely dysregulated in NAFLD.

---

[1] Department of Molecular and Integrative Physiology, University of Illinois at Urbana-Champaign, Urbana, IL 61801, USA. [2] Computational Biology Department, School of Computer Science, Carnegie Mellon University, Pittsburgh, PA 15213, USA. [3] Department of Pharmacology and Toxicology, Ernest Mario School of Pharmacy, Rutgers University, Piscataway, NJ 08854, USA. ✉email: jongsook@illinois.edu

Hepatic lipogenesis plays a critical role in the maintenance of whole-body energy homeostasis by the formation of triglyceride (TG)-rich very low-density lipoproteins and consequently, distribution of energy to non-hepatic tissues[1,2]. Hepatic lipogenesis is tightly regulated, mainly at the level of transcription, in response to nutritional states[1], but is elevated in obesity-induced non-alcoholic fatty liver disease (NAFLD) and diabetes[3,4]. After a meal, transcription of lipogenic genes is activated in response to elevated insulin and glucose levels by feeding-sensing factors, such as upstream stimulatory factors (USFs), sterol-regulatory element binding protein-1 (SREBP1), and carbohydrate-responsive element binding protein (CHREBP)[1,5,6]. While expression of lipogenic genes is substantially increased upon feeding, it is very low during fasting[1]. Whether the transition to low expression of lipogenic genes in the late fed state is simply a result of decreased activation or involves active repression of the genes is not known.

Fibroblast Growth Factor-15/19 (mouse FGF15, human FGF19) is a late fed-state gut hormone that is induced by the bile acid nuclear receptor, Farnesoid-X-Receptor (FXR/NR1H4), and functions to ensure a smooth metabolic transition from the fed to the fasted state[7–9]. FGF15/19 mediate postprandial hepatic responses, such as inhibition of bile acid and glucose synthesis and stimulation of protein and glycogen synthesis, independent of insulin action[10,11]. Indeed, FGF15/19 blood levels peak about 3 h after feeding[12–15], when blood insulin levels have returned to baseline. The overall metabolic actions of FGF15/19 resemble those of insulin, but in contrast to insulin, administration of FGF19 was shown to reduce both hepatic and plasma triglyceride (TG) levels in obese mice[7,8,16,17]. FGF19 mediates its postprandial functions in part via Small Heterodimer Partner (SHP/NR0B2). FGF19-mediated phosphorylation of SHP increases nuclear localization and gene repression activity of SHP, leading to transcription repression of bile acid synthesis, one-carbon cycle, and autophagy in the late fed state[18–22]. However, whether the FGF15/19 and FGF15/19-activated SHP have a physiological role in epigenetic repression of hepatic lipogenesis has not been shown.

Epigenetic modifications of DNA and histones link environmental signals to gene expression without changing the DNA sequence[23,24]. Although epigenetic regulation of hepatic lipogenic genes by histone modifying proteins has been studied[1,25–28], a role for DNA methyltransferases (DNMTs) has not been reported. In mammals, there are three major DNMTs, 1, 3A, and 3B[23]. While DNMT1 maintains stable 5-methylcytosine marks by methylating DNAs, DNMT3A and 3B are highly responsive to environmental cues and catalyze DNA methylation at CpG islands[23,24]. It is well known that aberrant DNA methylation is associated with numerous diseases[24,29–32]. Recently, obesity-induced hypermethylation at adiponectin and Fgf21 genes mediated by DNMT1 and DNMT3A, respectively, was shown to play an important role in the pathogenesis of insulin resistance in adipose tissue[29,30]. Although DNA methylation at hepatic lipogenic genes is low in NAFLD patients[31–34], the underlying mechanisms are unclear.

Here we report that FGF15/19 represses hepatic lipogenesis by activating SHP and DNMT3A physiologically, but this regulation is defective in obese mice and possibly in NAFLD patients. FGF15/19-activated SHP recruits DNMT3A to the Fasn promoter, resulting in DNA methylation and epigenetic gene repression. In NAFLD patients and obese mice, occupancy of SHP and DNMT3A and DNA methylation at the lipogenic genes are aberrantly low, which is consistent with elevated expression of these genes.

## Results

### Repression of hepatic lipogenic genes in the late fed state is dependent on both FGF15 and SHP.
As a first step in determining whether the lipogenic gene program is actively repressed in the late fed state, we examined the temporal expression of key lipogenic genes, fatty acid synthase (Fasn), sterol-regulatory element binding protein-1 (Srebp1), and acetyl-coA carboxylase-1 (Acc1), after refeeding C57BL/6 mice that had been fasted for 12 h. As expected, pre-mRNA levels, an indicator of transcription[35], of these genes quickly increased after refeeding, reaching a peak by about 1 h, but then rapidly decreased, returning to baseline levels by 6 h (Fig. 1a), whereas mRNA levels increased with a lag of 2 h and then, reached a plateau from 4 to 6 h (Fig. 1b). These results suggest that lipogenic genes are induced initially upon feeding but repressed later after feeding. The delayed and sustained increase in mRNA levels may in part reflect the time and efficiency of pre-mRNA processing, but also suggests the stability of mRNA is likely increased after feeding.

After a meal, FGF15/19 are secreted from the ileum and mediate postprandial metabolic responses in liver[7–9]. We, thus, examined whether FGF15 has a role in this late fed-state repression of lipogenic genes by examining Fasn as a model gene in FGF15-knockout (KO) mice. FASN is a key enzyme in de novo lipogenesis and its expression is primarily regulated at the level of transcription[1,25,26]. The rapid decrease in pre-mRNA levels of Fasn by 3 h after refeeding in control mice was blocked in the FGF15-KO mice (Fig. 1c, left). Fasn mRNA levels were significantly higher after 12 h of fasting and reached higher plateau levels at 6 h after refeeding in the KO mice compared to control littermates (Fig. 1c, right). Postprandial hepatic functions of FGF15/19 are mediated in part by SHP[10,21,22]. Similar to the results in FGF15-KO mice (Fig. 1c), the decrease in Fasn pre-mRNA levels in the late fed state was blocked in SHP-KO mice and plateau levels of the mRNA were increased in these mice (Fig. 1d). FGF15-KO and SHP-KO mice showed higher FASN protein levels after 12 h of fasting and also by 2 h after refeeding compared to control groups (Fig. 1e, f). These results demonstrate that the repression of Fasn in the late fed state is dependent on FGF15 and SHP, so that in the FGF15-KO or SHP-KO mice, both mRNA and protein levels increased more rapidly and reached higher levels.

### FGF19 globally represses lipogenic genes in which SHP binding sites overlap CpG islands.
To examine the role of FGF15/19 in repression of lipogenic genes, the global expression of genes in livers of mice treated with FGF19 was examined by RNA-seq. We used FGF19 since FGF15 is less stable, postprandial metabolic functions of these two enterokines are similar in general, and FGF19 has been utilized in previous mouse studies[10,11,20,22,36,37]. In FGF19-treated mice, 791 genes were downregulated, 951 were upregulated (Fig. 2a), and lipogenic genes were highly represented in the downregulated genes as analyzed by gene ontology (GO) (Fig. 2b). The repression of selected lipogenic genes by FGF19, as well as, the known FGF19 target bile acid synthetic gene, Cyp7a1[10,19,38,39], was confirmed by RT-qPCR (Fig. 2c).

In SHP ChIP-seq studies of livers from FGF19-treated mice[18], CpG islands, which could potentially be methylated, were near about 80% of the SHP cistrome (Fig. 2d). Indeed, in selected lipogenic genes, SHP peaks were present at promoter regions that contain CpG islands (Fig. 2e), suggesting the possibility that FGF19-induced SHP binding at lipogenic genes may correlate with DNA methylation and gene repression.

Consistent with this prediction, in liver ChIP and methylated DNA IP (MeDIP) assays, SHP occupancy and DNA methylation were both increased at the promoter regions of lipogenic genes by FGF19 treatment, but only SHP occupancy was increased at Cyp7a1, a well-known SHP target gene (Fig. 2f, g). The increase in methylation at Fasn was confirmed at specific sites by bisulfite

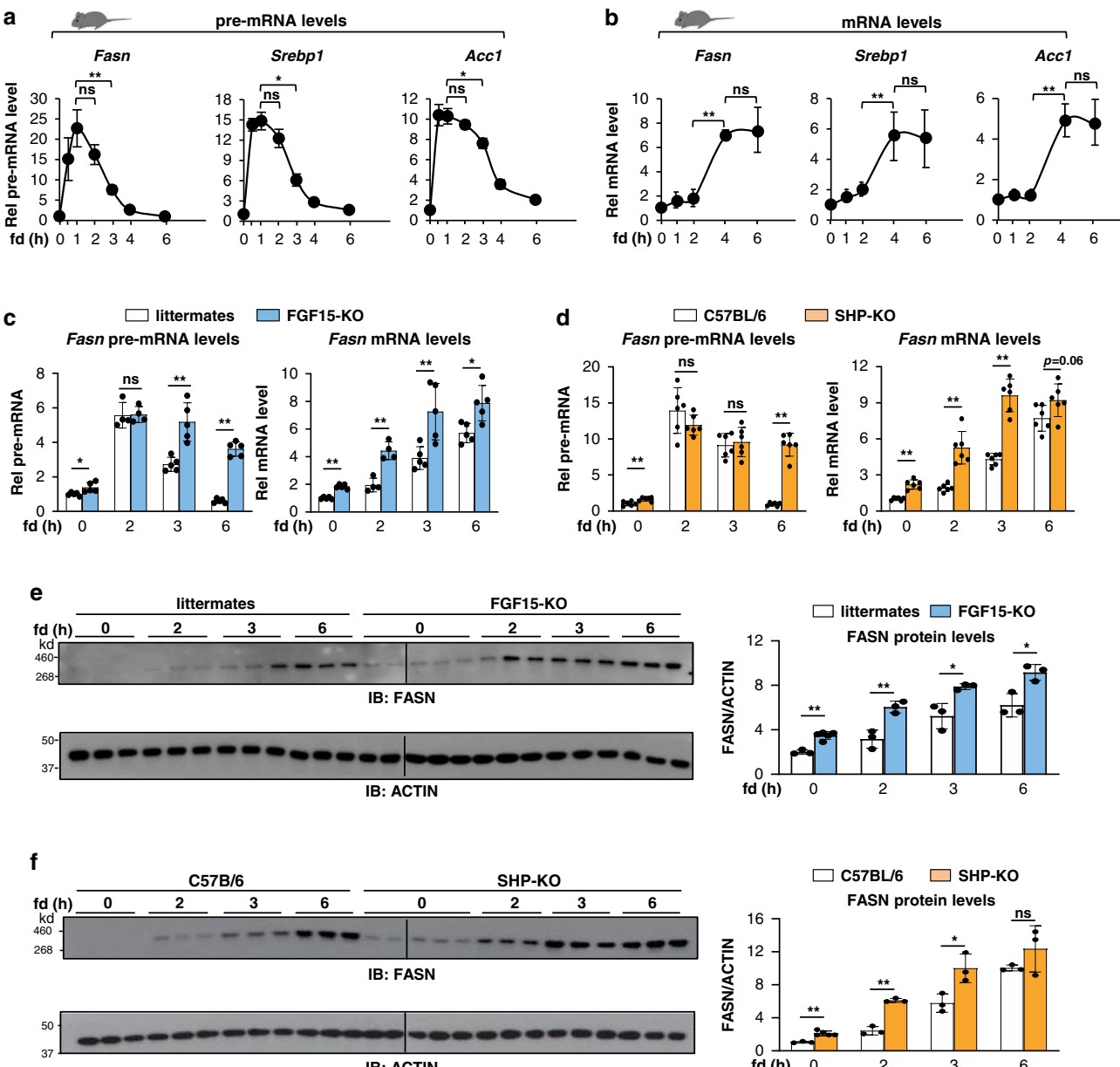

**Fig. 1 Late fed-state transcriptional repression of lipogenic genes is dependent on both FGF15 and SHP.** Mice were fasted overnight and refed normal chow for the indicated times. **a** Relative levels of pre-mRNAs of lipogenic genes in the livers of C57BL/6 mice measured by RT-qPCR using primers spanning the intron 1 and exon 1 boundaries and **b** mRNA levels of these genes. **c–f** Relative levels of hepatic *Fasn* pre-mRNA (**c**, left), mRNA (**c**, right), and protein (**e**) in FGF15-KO mice or control littermates. Relative levels of hepatic *Fasn* pre-mRNA (**d**, left), mRNA (**d**, right), and protein (**f**) in SHP-KO mice or control C57BL/6. Data presented as mean values ± Standard Deviation (SD) are shown (**a**, **b**, $n = 5$ mice; **c**, $n = 4$ for 2 h, $n = 5$ for other times, mice; **d**, $n = 6$ mice; **e**, **f**, $n = 5$ for 0 time, FGF15-KO mice, $n = 3$ for other times, mice), and statistical significance was determined by (**a**, **b**) one-way ANOVA with the Tukey post-test or (**c–f**) two-tailed Student's *t*-test. * $P < 0.05$, ** $P < 0.01$, ns, statistically not significant.

sequencing (Fig. 2h). These results suggest that repression of a subset of direct SHP targets, including lipogenic genes but not *Cyp7a1*, involve DNA methylation. Indeed, FGF19 treatment increased DNA methylation at the *Fasn* promoter (Fig. 2i) and resulted in decreased mRNA levels of *Fasn* in control, but not in SHP-KO mice (Fig. 2j). Together, these results show that FGF19-mediated repression of *Fasn* is dependent on SHP and suggest that the repression is, in part, via DNA methylation.

**Recruitment of DNMT3A and DNA methylation at the *Fasn* promoter are dependent on both FGF19 and SHP.** Since DNA methylation is catalyzed by a family of highly conserved DNMTs

that include DNMT1, 3A, and 3B[23], we examined whether SHP interacts with the DNMTs and whether FGF19 affects this interaction.

FGF19 treatment dramatically increased the interaction of SHP with DNMT3A and, to a lesser extent with DNMT3B, while interaction with DNMT1 was not detected (Fig. 3a). Consistent with these results, in ChIP assays, FGF19 treatment substantially increased the occupancy of DNMT3A at the *Fasn* promoter, while statistically significant increases of DNMT3B or DNMT1 occupancy were not observed (Fig. 3b, left). Consistent with undetectable DNA methylation at *Cyp7a1* (Fig. 2g), FGF19 did not increase occupancy of any of the DNMTs at the *Cyp7a1* promoter (Fig. 3b, right). In addition to *Fasn*, FGF19 increased

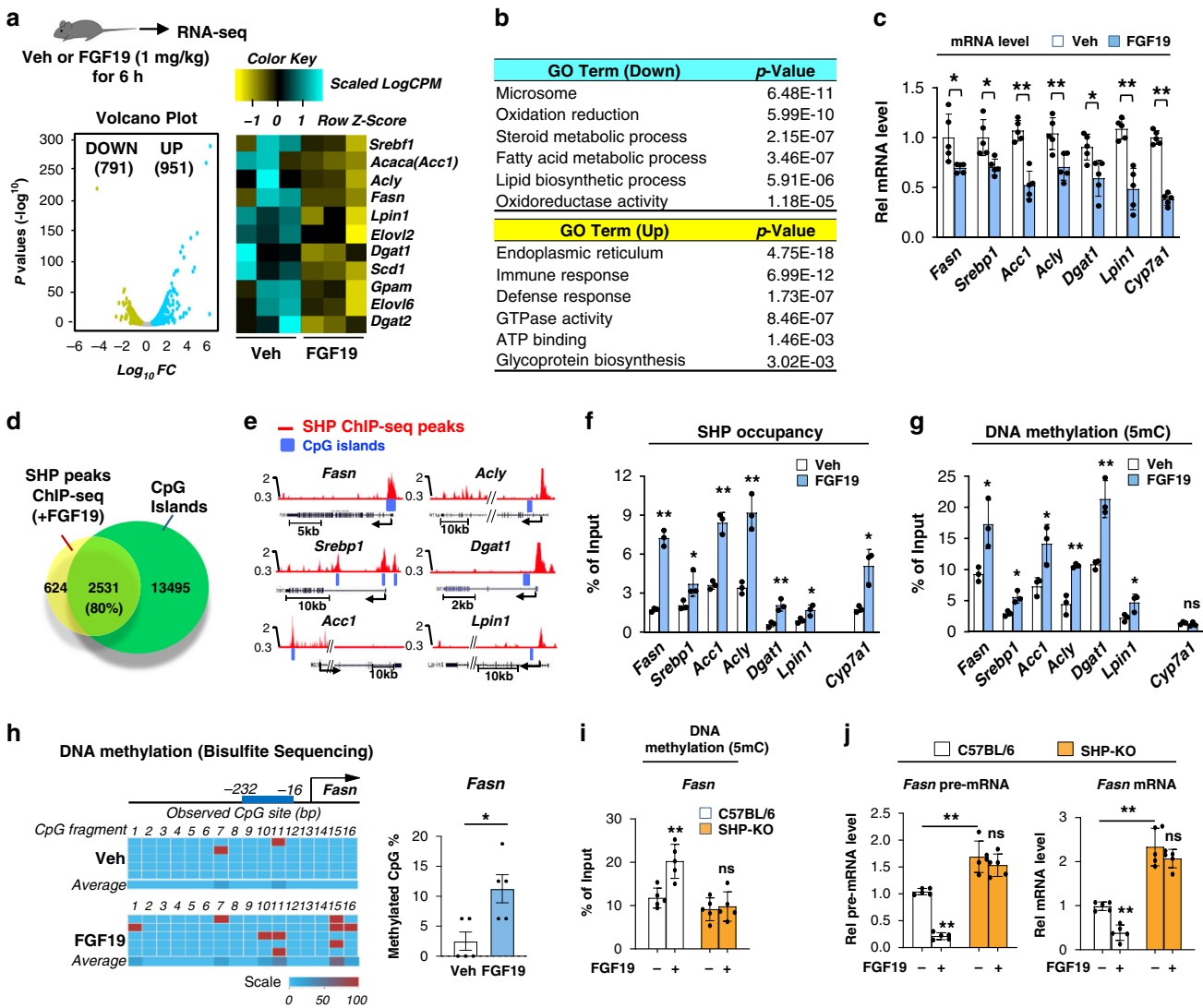

**Fig. 2 SHP occupancy and DNA methylation are increased at lipogenic genes after FGF19 treatment. a–c** C57BL/6 mice were treated with vehicle or FGF19 for 6 h after fasting overnight. **a, b** Total RNA was isolated from mouse liver and analyzed by RNA-seq. **a** A volcano plot showing genome-wide changes in mRNA level (left) and a heatmap of the differential expression of lipogenic genes in mice treated with vehicle or FGF19 (right). **b** Selected biological pathways increased or decreased by FGF19 treatment as determined by gene ontology (GO) analysis. **c** Relative mRNA levels of selected lipogenic genes measured by RT-qPCR. **d** Venn diagrams showing the numbers of SHP binding peaks increased by FGF19 treatment of mice for 2 h in hepatic chromatin using published ChIP-seq data[18] and the numbers of CpG islands. **e** Normalized SHP binding peaks (Red) and CpG islands (Blue) at selected lipogenic genes displayed using the UCSC genome browser. **f, g** C57BL/6 mice were treated with vehicle or FGF19 for 2 h. **f** SHP occupancy or **g** DNA methylation at promoters of the indicated lipogenic genes determined by ChIP or MeDIP, respectively. **h** DNA methylation at specific sites of *Fasn* promoter determined by bisulfite sequencing. **i, j** C57BL/6 and SHP-KO mice were treated with vehicle or FGF19 for 2 h (**i, j**-left) or 6 h (**j**-right). **i** DNA methylation determined by MeDIP at hepatic *Fasn* and **j** hepatic pre-mRNA and mRNA levels. **c, f-j** The mean and standard deviation are plotted (*n* = 5). Data presented as mean values ± SD are shown (**c, h-j**, *n* = 5 mice; **f, g**, *n* = 3 mice), and statistical significance was determined by (**c, f-h**) two-tailed Student's *t*-test or (**i, j**) two-way ANOVA with the Tukey post-test. *$P < 0.05$, **$P < 0.01$, ns, statistically not significant. In **b**, exact p-values are produced by DAVID v6.8 program (https://david.ncifcrf.gov/) which are computed by summing probabilities p over defined sets of tables (Prob = $\sum Ap$).

DNMT3A occupancy at the lipogenic genes, *Srebp1*, *Dgat1*, and *Acc1* (Supplementary Fig. 1). We, thus, focused our studies on DNMT3A.

To examine if SHP is required for the FGF19-induced recruitment of DNMT3A to the *Fasn* promoter, we utilized SHP-KO mice. The FGF19-mediated increase in DNMT3A occupancy at the *Fasn* promoter was largely abolished in SHP-KO mice (Fig. 3c, left). In contrast, FGF19 did not increase DNMT3A occupancy at *Cyp7a1* (Fig. 3c, right). SHP represses genes by recruiting LSD1 histone demethylase[19,39,40]. Indeed, LSD1 binding at *Fasn* was decreased and H3K4-me3 levels were increased in SHP-KO mice compared to C57BL/6 mice

(Supplementary Fig. 2). These results indicate that FGF19-induced recruitment of DNMT3A to *Fasn* is SHP-dependent.

**FGF15/19-induced DNA methylation at *Fasn* occurs in the late fed state.** Next, we determined whether the FGF15/19-induced DNA methylation occurs physiologically in the late fed state. Since blood FGF15/19 levels peak around 3 h after feeding[12–15], we examined the interaction of SHP with DNMT3A and DNA methylation in mice that were refed for 3 h after fasting overnight.

Refeeding increased the interaction between SHP and DNMT3A in control littermates, but not in FGF15-KO mice (Fig. 3d). Consistent with these results, feeding increased CpG DNA

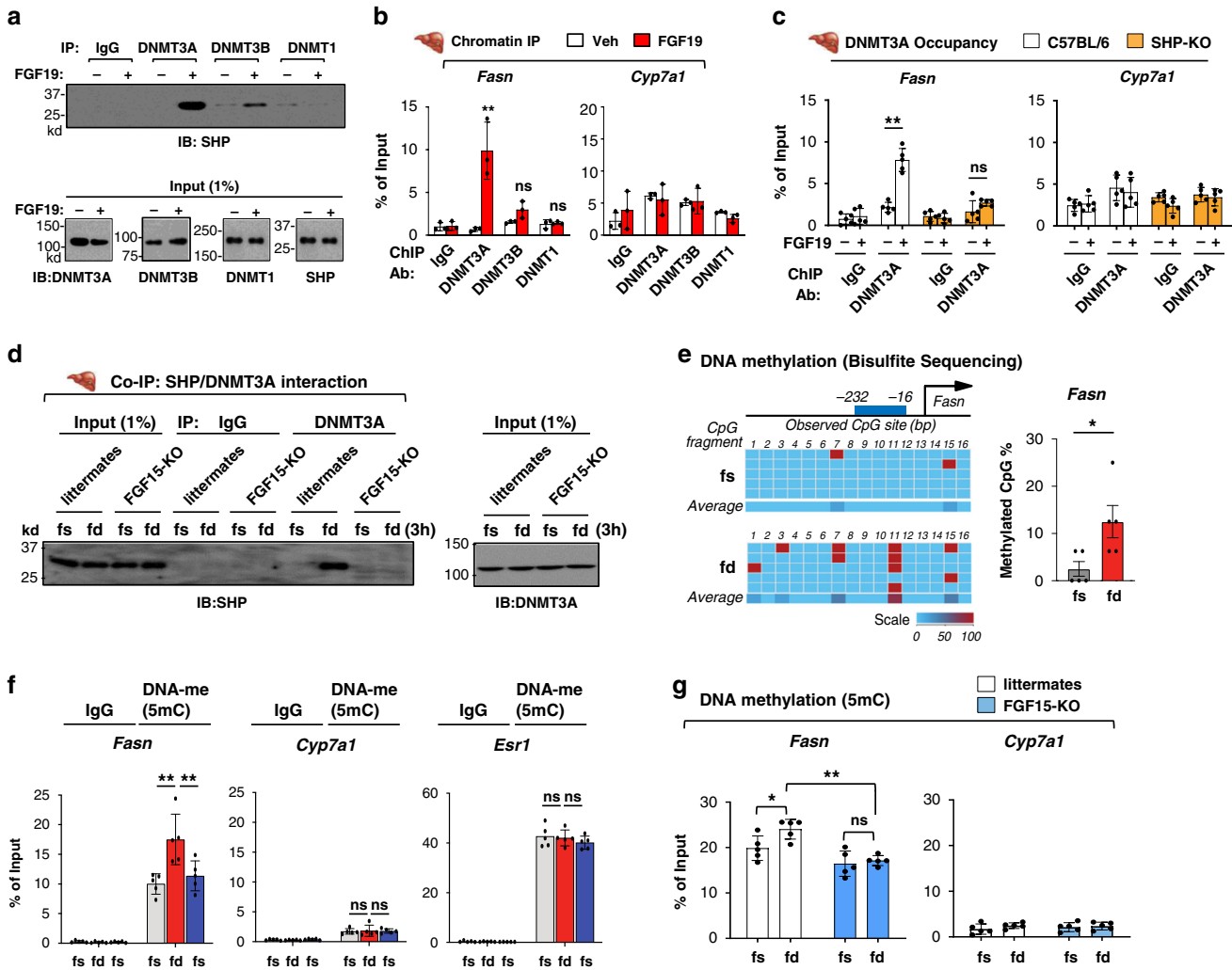

**Fig. 3 Increased DNMT3A occupancy and DNA methylation at the *Fasn* promoter in the late fed state are dependent on both SHP and FGF15/19.**
**a**–**c** Mice were treated with vehicle or FGF19 for 2 h after fasting overnight. **a** Interaction of SHP with DNMTs in liver extracts determined by CoIP.
**b** Occupancy of DNMTs at *Fasn and Cyp7a1* determined by ChIP. **c** Occupancy of DNMT3A at the *Fasn* and *Cyp7a1* determined by ChIP in SHP-KO or
C57BL/6 mice. **d** FGF15-KO mice or littermates were fed normal chow for 3 h (fd) after fasting overnight (fs). Interaction of DNMT3A and SHP in liver
extracts determined by CoIP (left). Input SHP in the liver determined by IB of 1% of the liver extract used for the CoIP assays (right). Tissues from 3 mice
were pooled for each sample and the experiment was done twice. **e**–**f** C57BL/6 mice were fed normal chow for 3 h (fd) after fasting overnight (fs) and in
(**f**) were fasted again overnight. **e** DNA methylation at specific sites of *Fasn* promoter determined by bisulfite sequencing. **f** DNA methylation at the *Fasn,
Cyp7a1, and Esr1* promoters determined by MeDIP. **g** FGF15-KO mice or littermates were fed normal chow for 3 h (fd) after fasting overnight (fs). DNA
methylation at *Fasn* and *Cyp7a1* determined by MeDIP. **b**, **c**, **f**, **g** Mean values ± SD are plotted (**b**, n = 3 mice; **c**, **e**, **f**, **g**, n = 5 mice). Statistical significance
was determined by (**b**, **c**, **f**, **g**) two-way ANOVA with Tukey post-test or (**e**) two-tailed Student's *t* test. *P < 0.05, **P < 0.01, ns, statistically not significant.

methylation at the *Fasn* promoter detected by bisulfite sequencing (Fig. 3e) or by MeDIP (Fig. 3f, left). The increase in CpG methylation at *Fasn* after refeeding returned to the basal levels after fasting (Fig. 3f, left), suggesting that reversible DNA methylation occurs at the *Fasn* promoter during a full fasting/feeding/fasting cycle. In contrast, DNA methylation at *Cyp7a1* was not detected (Fig. 3f, center) and DNA methylation at a highly methylated gene, *Esr1*[41], was not changed (Fig. 3f, right). Further, feeding-mediated increases in DNA methylation at *Fasn* was not detected in FGF15-KO mice (Fig. 3g). These results indicate that increased levels of FGF15 in the late fed state induces the SHP interaction with DNMT3A and increases DNA methylation at the *Fasn* promoter, which is associated with gene repression.

**Liver-specific downregulation of DNMT3A in mice partially reverses FGF19-mediated repression of lipogenesis.** To examine

the functional significance of hepatic DNMT3A in epigenetic repression of lipogenesis by FGF15-SHP, DNMT3A was down-regulated by AAV8-mediated expression of shRNA (Fig. 4a). DNMT3A protein levels were downregulated specifically in the liver, but not in intestine or adipose tissue (Fig. 4b). Expression of nearly all of the lipogenic genes tested, was increased after downregulation of DNMT3A, while that of *Cyp7a1* was not (Fig. 4c).

FGF19 treatment increased DNA methylation at the *Fasn* promoter (Fig. 4d) and decreased pre-mRNA and mRNA levels of *Fasn* (Fig. 4e, f), and these FGF19-mediated effects were diminished in DNMT3A-downregulated mice (Fig. 4d–f). Consistent with these results, downregulation of hepatic DNMT3A markedly increased basal de novo lipogenesis (Fig. 4g) and hepatic TG levels (Fig. 4h) and significant inhibition of lipogenesis and TG levels by FGF19 was not observed after downregulation of DNMT3A. These results indicate that FGF19-

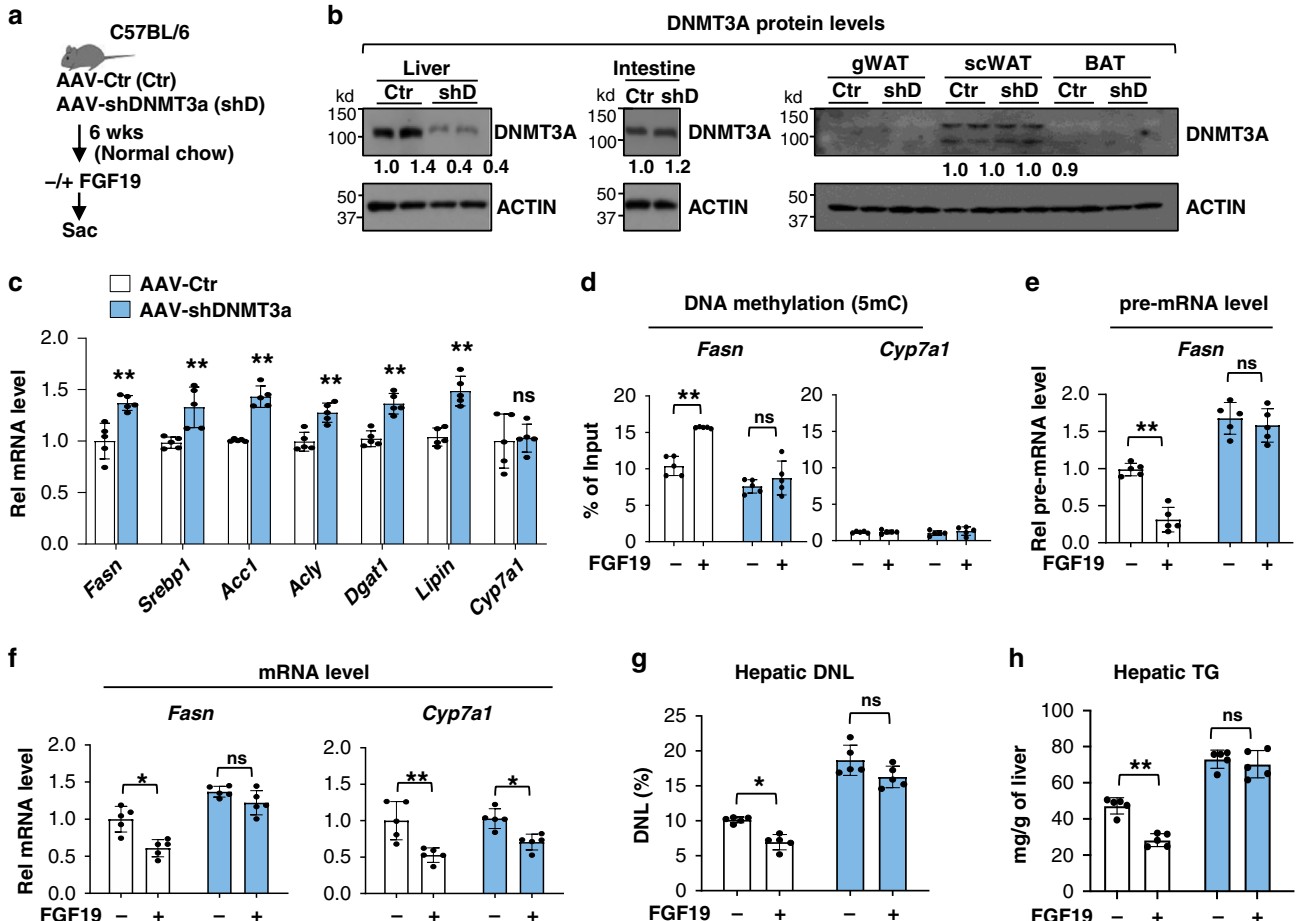

**Fig. 4 FGF19-mediated DNA methylation and repression of hepatic lipogenesis are attenuated by liver-specific downregulation of DNMT3A.**
**a** Experimental outline. C57BL/6 mice were infected with AAV-shDNMT3A (blue bars) or AAV-empty (Ctr, white bars) and 6 weeks later, fasted for 12 h and then treated with FGF19 or vehicle for 2 h for MeDIP and 2 h or 6 h for RT-qPCR detecting pre-mRNA or mRNA, respectively. **b** DNMT3A expression in the liver, intestine, and adipose tissues determined by IB. **c** The mRNA levels of lipogenic genes determined by RT-qPCR. **d** DNA methylation at the *Fasn* and *Cyp7a1* promoters determined by MeDIP. Samples from two mice for each group are shown. The experiment was done once. **e**, **f** *Fasn* pre-mRNA and mRNA levels and *Cyp7a1* mRNA levels determined by RT-qPCR. **g**, **h** C57BL/6 mice were infected with AAV-shDNMT3A or AAV-empty (Ctr) and 6 weeks later were treated daily with vehicle or FGF19 and given 4% deuterated drinking water for 2 days. Hepatic de novo lipogenesis (DNL) and TG levels measured as described in Methods. **c–h** The mean values ± SD are plotted ($n = 5$ mice). Statistical significance was determined by (**c**) two-tailed Student's $t$ test or (**d-h**) two-way ANOVA with the Tukey post-test. *$P < 0.05$, **$P < 0.01$, ns, statistically not significant.

induced DNA methylation at the *Fasn* promoter is largely dependent on hepatic DNMT3A and that the DNA methylation correlates with the decreased *Fasn* expression and lipogenesis.

**SHP recruits DNMT3A to the SREBP1-transactivated *Fasn* gene in the late fed state.** We next determined how SHP and DNMT3A mediate transcriptional repression of lipogenic genes in response to the FGF15/19 signal. Since SHP does not directly bind to DNA[18–22], we first identified potential transcription factors with which SHP might interact. Numerous factors, such as SREBP1, CHREBP, and USFs, activate *Fasn* gene expression in response to feeding[1,5,6]. Since SREBP1 is a key lipogenic activator, and in published ChIP-seq studies[18], sterol response element motifs were present most frequently within the global SHP cistrome, we examined the interactions of SREBP1 with SHP and DNMT3A in transcriptional regulation of the *Fasn* gene.

Consistent with the finding that blood levels of FGF15/19 peak around 3 h after feeding[12–15], refeeding for 1 h after fasting did not increase the interaction between SHP and DNMT3A, but the interaction was increased late after feeding, reaching a peak at 4 h (Fig. 5a). Refeeding for 4 h dramatically increased the interaction

of SREBP1 with DNMT3A, as well as with SHP (Fig. 5b). In liver ChIP assays, occupancy of SREBP1 at the *Fasn* promoter was increased as early as 1 to 2 h after refeeding and the increase was sustained up to 6 h (Fig. 5c). In contrast, occupancy of SHP and DNMT3A was delayed with little increase at 2 h, followed by substantial increases at 4 h after feeding (Fig. 5c). In FGF15-KO mice, binding of DNMT3A and SHP did not increase, whereas the increase in SREBP1 binding was unchanged (Fig. 5d). These results indicate that physiologically induced FGF15 in the late fed state is required for binding of DNMT3A and SHP at the *Fasn* promoter. FGF19 treatment increased binding for both SHP and DNMT3A to SREBP1-bound chromatin (Fig. 5e), suggesting that these factors co-occupy the *Fasn* promoter.

To examine the functional role of the binding of SREBP1, SHP, and DNMT3A, to the *Fasn* promoter, we examined the effects of these factors on the activity of a *Fasn* promoter-luciferase reporter. SREBP1 increased *Fasn* promoter activity and the increase was inhibited by SHP (Fig. 5f). Further, co-expression of DNMT3A wild type (WT) with SHP enhanced inhibition of the *Fasn* promoter activity by SHP, but co-expression with a catalytically inactive DNMT3A-C706S mutant[29] had little effect (Fig. 5f). These results suggest that the DNA methyltransferase

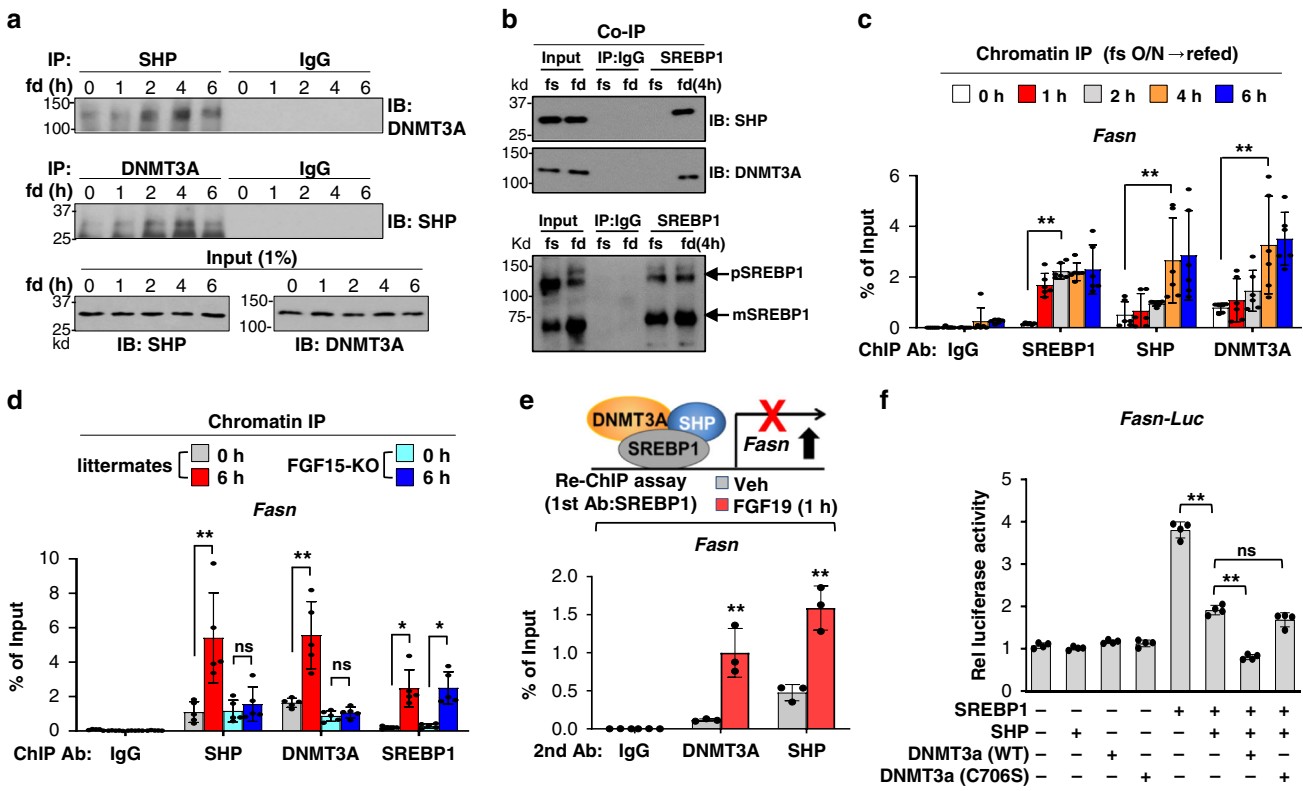

**Fig. 5 SHP and DNMT3A inhibit SREBP1-mediated induction of *Fasn* in the late fed state. a–c** C57BL/6 mice were fed (fd) for the indicated times after fasting overnight (fs). Interaction of (**a**) DNMT3A with SHP or of (**b**) SREBP1 with SHP or DNMT3A in liver extracts determined by CoIP. **c** Occupancy of SREBP1, SHP, and DNMT3A at the *Fasn* promoter determined by ChIP. **d** FGF15-KO mice or littermates were fed for 6 h after overnight fasting. Occupancies of SHP, DNMT3A, or SREBP1 at the hepatic *Fasn* promoter determined by ChIP. **e** C57BL/6 mice were treated with FGF19 for 1 h after fasting overnight. Co-occupancy of SHP or DNMT3A with SREBP1 at the *Fasn* promoter determined by sequential re-ChIP. **f** PMHs were transfected with plasmids as indicated for 48 h and then, treated with FGF19 (50 ng/ml) or vehicle for 6 h. Luciferase activity normalized to β-galactosidase activity. **c–f** Mean values ± SD are plotted (**c**, *n* = 6 mice; **d**, *n* = 5 mice, **e**, *n* = 3 mice, **f**, *n* = 4 mice). Statistical significance was determined by (**c–e**) two-way ANOVA or (**f**) one-way ANOVA with the Tukey post-test. \**P* < 0.05, \*\**P* < 0.01, ns, statistically not significant.

activity of DNMT3A is important for its enhanced repression of the *Fasn* promoter activity by SHP. These findings are consistent with the early activation after feeding of *Fasn* by insulin-activated SREBP1[1,5], followed by recruitment of SHP and DNMT3A in the late fed state, which leads to epigenetic repression of the *Fasn* gene expression.

**FGF15/19-induced phosphorylation of SHP is required for its interaction with DNMT3A and inhibition of hepatic lipogenesis.** Since FGF15/19 signaling-induced phosphorylation of SHP at Thr-55 (mouse Thr-58) increases its nuclear localization and gene repression function[20,38,39], we next tested whether this phosphorylation is important for repression of lipogenic genes in C57BL/6 mice (Fig. 6a). Adenoviral-mediated expression of SHP-WT resulted in increased DNA methylation at *Fasn* and *Srebp1* promoters (Fig. 6b) and decreased expression of these genes (Fig. 6c). These changes, however, were not observed with expression of the T55A mutant, and in fact, expression of the genes was increased and DNA methylation was decreased compared to the Ad-empty control (Fig. 6b, c), which is consistent with the dominant-negative characteristics of this mutant noted previously[18,21,39]. Similarly, expression of SHP-WT reduced hepatic lipogenesis and liver TG levels and neutral lipids, while expression of the T55A mutant had the reverse effects (Fig. 6d, e, Supplementary Fig. 3). These results demonstrate that FGF15/19-induced phosphorylation of SHP is required for repression of the lipogenesis and decreased TG levels in the liver.

To avoid confounding effects of endogenous SHP expressed in C57BL/6 mice in the studies above and to further test whether the SHP phosphorylation is important for repression of *Fasn* in a cell-autonomous manner, expression of SHP-WT, p-defective T55A-SHP, or p-mimic T55D-SHP was reconstituted in primary hepatocytes isolated from SHP-KO mice. FGF19 treatment increased the interaction of DNMT3A with SHP-WT, while the T55A mutation decreased the interaction and the T55D mutation increased the interaction even in cells not treated with FGF19 (Fig. 6f). Further, in reporter assays in hepatocytes treated with FGF19, co-expression of DNMT3A with SHP-WT or T55D-SHP, but not with T55A-SHP, decreased the SREBP1-mediated *Fasn* promoter activity (Fig. 6g). Together, these results indicate that FGF15/19-induced phosphorylation of SHP is required for its interaction with DNMT3A in repression of *Fasn* and hepatic lipogenesis.

**Liver-specific downregulation of DNMT3A partially reverses SHP-mediated inhibition of lipogenesis and hepatosteatosis in obese mice.** Adenoviral-mediated liver-specific expression of SHP was shown to reduce hepatic TG levels in obese mice[21,39]. We, thus, examined whether the repression of lipogenic genes via DNA methylation and consequently, decreased hepatic lipogenesis, played a role in the SHP-mediated lipid-lowering effects, and further examined the role of DNMT3A in mediating these effects. Mice were fed with either a normal chow diet (ND) or a high-fat and high-fructose (HF/HF) diet and then, SHP was expressed

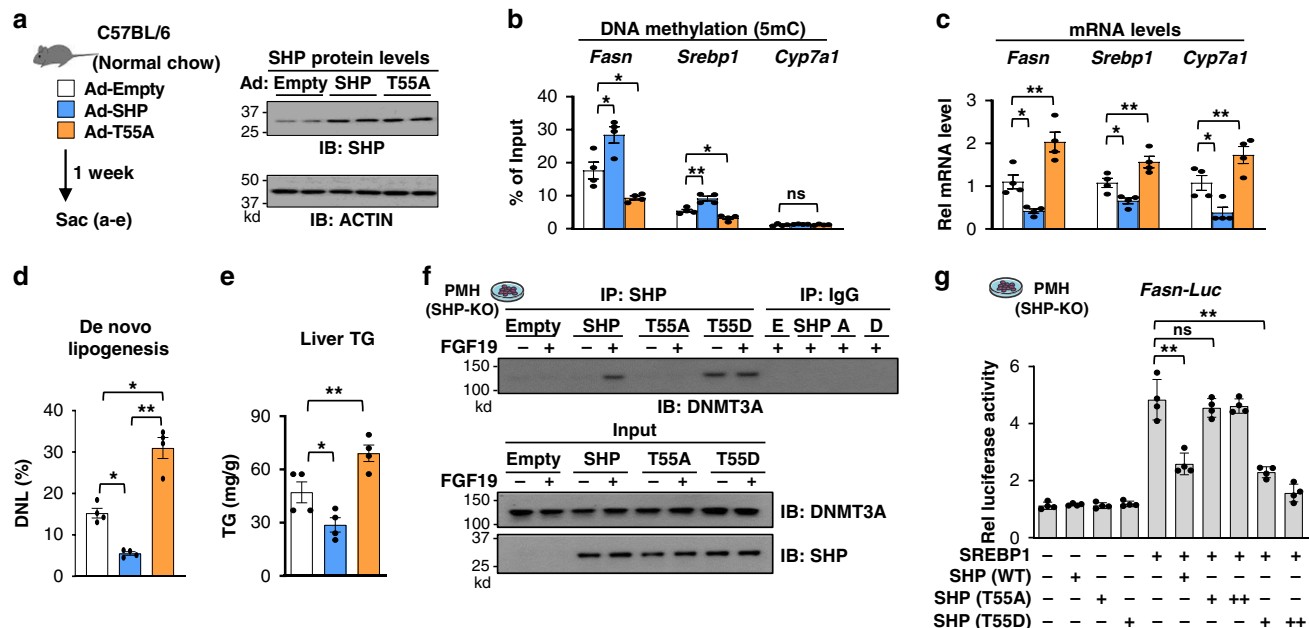

**Fig. 6 FGF15/19-induced phosphorylation of SHP is required for its interaction with DNMT3A in repression of hepatic lipogenesis. a–e** C57BL/6 mice were infected by adenoviruses that express SHP-WT or T55A or by empty adenoviruses, and one week later the mice were sacrificed. **a** Experimental outline (left). Levels of SHP protein determined by IB (right). **b** DNA methylation at the promoters of the indicated genes measured by MeDIP. **c** Levels of mRNA of the indicated genes measured by RT-qPCR. **d** For 2 days prior to brief fasting and sacrifice, drinking water for the mice was 4% deuterated water. Hepatic DNL was measured as described in Methods. **e** Liver triglyceride (TG) levels. **f–g** PMHs were prepared from SHP-KO mice and cultured for 48 h. The PMHs were transfected with plasmids expressing control, SHP-WT, a phosphorylation-defective SHP-T55A mutant, or a phosphorylation-mimic SHP-T55D mutant. Seventy two hours later, cells were cultured in serum-free media overnight and then treated with vehicle or FGF19 for 2 h (**f**) or 6 h (**g**). **f** Interaction of SHP and DNMT3A determined by CoIP assays. Four culture plates were pooled for each sample. The experiment was repeated two times with similar results. **g** Luciferase activities normalized to β-galactosidase activities in PMH transfected with the indicated plasmids. **b–e, g** The mean values ± SD are plotted (**b–e**, n = 4 mice; **g**, n = 4 PMH culture dishes). Statistical significance was determined by one-way ANOVA with the Tukey post-test. *P < 0.05, **P < 0.01, ns, statistically not significant.

and/or DNMT3A was downregulated in the liver using viral expression vectors (Fig. 7a, b).

Liver-specific overexpression of SHP decreased body weight without changes in food intake (Fig. 7c), substantially reduced the size of liver in the obese mice near to that of mice fed a normal chow, and also decreased the ratio of liver/body weight (Fig. 7d). Liver-specific downregulation of DNMT3A in obese mice at least partially blocked each of these changes mediated by SHP overexpression, while the effects of downregulation of DNMT3A alone were not significantly different from those in mice infected with AAV-empty (Fig. 7c, d). Liver neutral lipids detected by H&E and Oil Red O staining and TG levels in liver and plasma were markedly decreased (Fig. 7e, f) and glucose tolerance was also improved after overexpression of SHP (Fig. 7g). Each of these SHP-mediated beneficial effects was reversed, at least partially, by downregulation of hepatic DNMT3A (Fig. 7c–g).

In mechanistic studies, SHP expression increased DNA methylation at the *Fasn* and *Srebp1* promoters (Fig. 7h), decreased mRNA levels (Fig. 7i) and decreased lipogenesis (Fig. 7j), and each of these SHP-mediated effects was blunted by downregulation of DNMT3A (Fig. 7h–j). These results demonstrate that SHP-mediated repression of hepatic lipogenesis and amelioration of fatty liver in obese mice is largely dependent on DNMT3A.

**In obese mice, decreased occupancy of SHP and DNMT3A and DNA methylation at *Fasn* are associated with increased gene expression.** Previous studies have suggested that FGF15/19 signaling is defective in fatty livers of obese mice and NAFLD patients[21,42,43]. Because FGF15/19 enhances the gene repression

function of SHP[21,38], we examined whether the occupancy of SHP and DNMT3A and DNA methylation levels at the *Fasn* promoter are altered in obese mice.

Occupancy of SHP and DNMT3A was significantly reduced at the *Fasn* promoter and DNA methylation was decreased at *Fasn and Srebp1* promoters in obese mice fed high-fat diet (HFD) (Fig. 8a, b). Consistent with these results, mRNA levels of these genes were increased over 4-fold in obese mice (Fig. 8c) and protein levels of FASN, detected only in the cytoplasm, and SREBP1, detected primarily in the nucleus, were also increased (Fig. 8d) as described previously[44]. DNMT3A protein levels, predominantly in the nucleus, were similar in the lean and obese mice (Fig. 8d). While total SHP levels were not markedly changed in the obese mice, intriguingly, SHP nuclear levels were substantially reduced (Fig. 8d). The decreased nuclear SHP levels and SHP binding at the *Fasn* promoter in obese mice are consistent with decreased binding of DNMT3A and DNA methylation, and increased *Fasn* expression.

**In NAFLD patients, lipogenic gene expression is increased, which is consistent with decreased occupancy of SHP and DNMT3A and DNA methylation.** Expression of lipogenic genes are highly elevated and DNA methylation at lipogenic genes is aberrantly low in NAFLD patients[3,4], but the underlying mechanisms are unclear. We, thus, examined whether the occupancy of SHP and DNMT3A and DNA methylation at lipogenic genes are altered in NAFLD patients with steatosis or NASH-fibrosis.

The mRNA levels of key lipogenic genes, *FASN, SREBP1* and *DGAT1*, were substantially increased and those of DNMT3A were

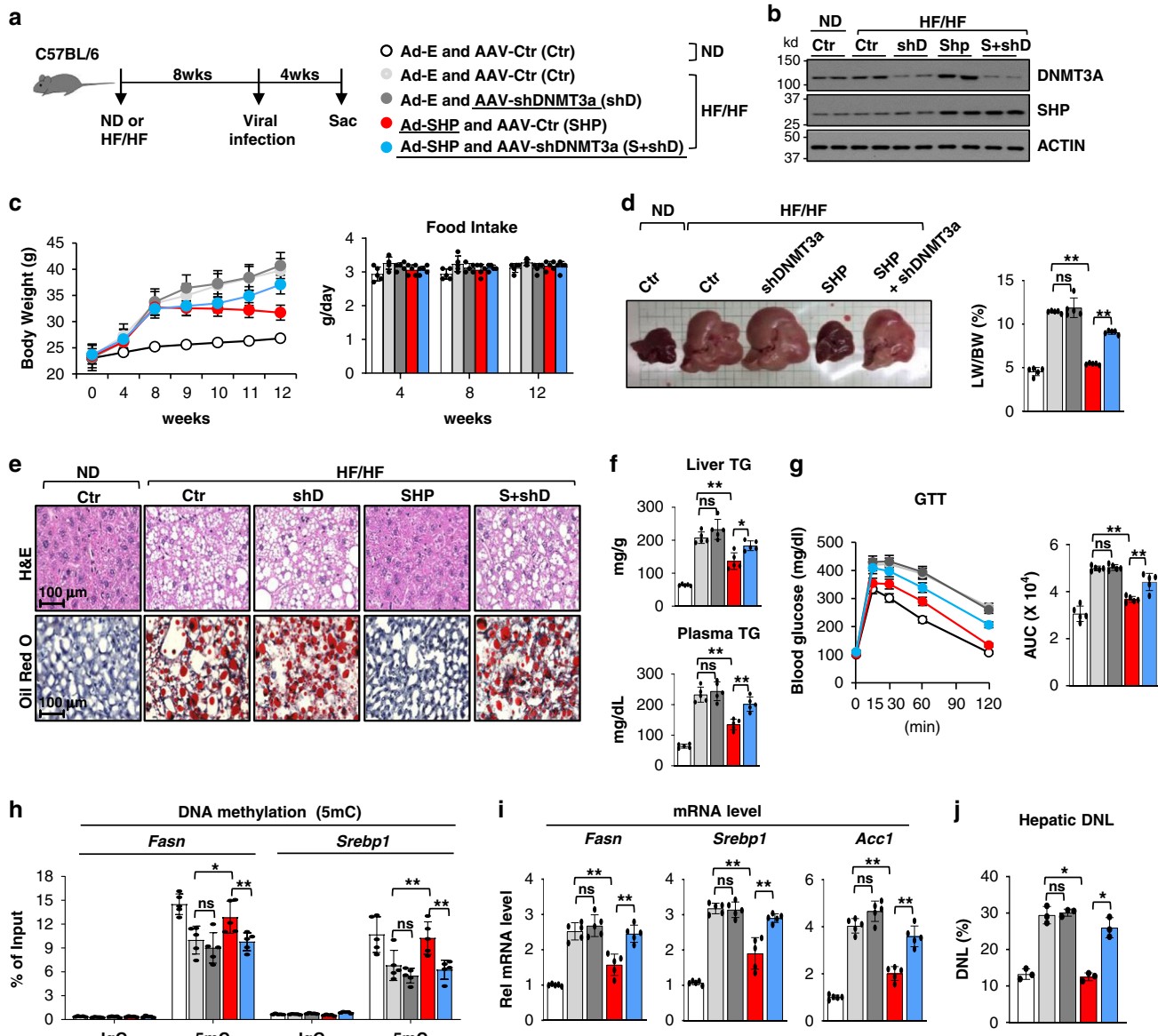

**Fig. 7 SHP-mediated repression of lipogenesis and amelioration of fatty liver in obese mice are diminished by liver-specific downregulation of DNMT3A.** C57BL/6 mice were fed normal chow diet (ND) or a high-fat /high-fructose (HF/HF) diet for 8 weeks, then infected with the combinations indicated in (**a**) of AAV-empty (Ctr), AAV-shDNMT3A, Ad-SHP, and Ad-empty (Ctr) and sacrificed 4 weeks later. **a** Experimental outline. **b** Levels of DNMT3A and SHP protein in the liver determined by IB. Shown are results for two mice and IB analysis was done twice. **c** Changes in body weight (left) and food intake (right). **d** Representative images of livers tissues (left) and the ratio of liver/body weight (right, $n = 5$). **e** Liver sections stained with H&E (upper panels) and Oil Red O (lower panels). Representative images are shown for one mouse in each group. Similar results were observed in two additional mice in each group. **f** Liver (upper) and plasma (lower) TG levels. **g** Glucose tolerance test (GTT) with calculated areas under the curve (AUC) plotted (right). **h** DNA methylation at *Fasn* and *Srebp1* promoter regions measured by MeDIP. **i** Levels of mRNAs of the indicated lipogenic genes determined by RT-qPCR. **j** Mice were given 4% deuterated drinking water for 2 days and DNL was measured as described in methods. **c, d, f–j** The mean values ± SD are plotted (**c, d, f–i**, $n = 5$ mice; **j**, $n = 3$ mice). Statistical significance was determined by (**d, f, g, i, j**) one-way ANOVA or (**h**) two-way ANOVA with the Tukey post-test. *$P < 0.05$, **$P < 0.01$, ns, statistically not significant.

modestly increased, while *SHP* mRNA levels were little changed in the NAFLD patients (Fig. 9a). The protein levels of *FASN* were also elevated in the patients with steatosis and further elevated in the patients with NASH-fibrosis, whereas protein levels of both DNMT3A and SHP were not significantly changed in these patients compared to normal subjects (Fig. 9b). Despite unchanged proteins levels of DNMT3A and SHP in the patients, the occupancy of DNMT3A and SHP at the lipogenic genes, *FASN, SREBP1 and DGAT1*, was decreased, which is consistent with decreased DNA methylation (Fig. 9c, d, Supplementary Figs. 4, 5). In contrast, DNA methylation at *CYP7A1* was not

detectable in either normal subjects or the NAFLD patients (Supplementary Fig. 5). These results suggest that decreased occupancy of DNMT3A and decreased DNA methylation at lipogenic genes contribute to elevated gene expression in the NAFLD patients.

## Discussion

Hepatic lipogenesis is normally tightly regulated, increasing in response to elevated blood insulin and glucose levels upon feeding and decreasing during fasting[1,2,6]. Dysregulation of this control

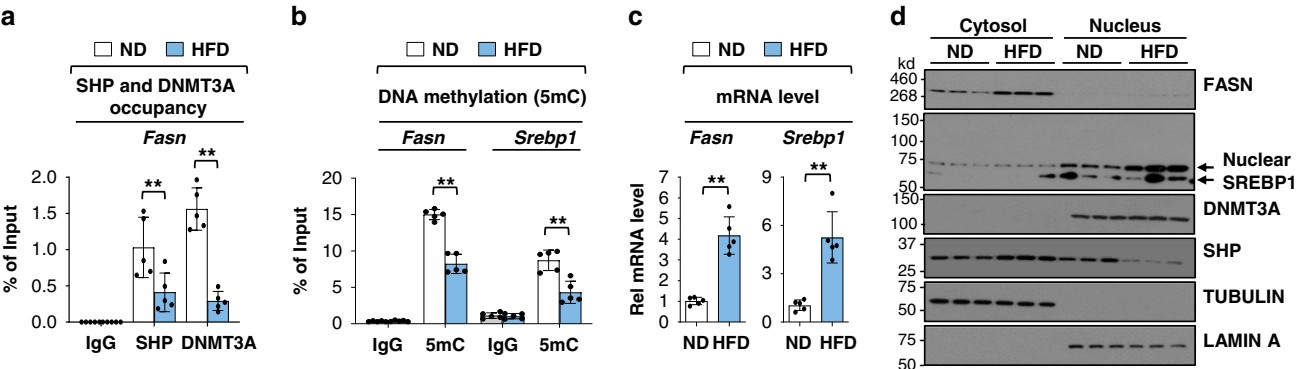

**Fig. 8 In obese mice, occupancy of SHP and DNMT3A and DNA methylation at lipogenic genes are low, with elevated gene expression.** C57BL/6 mice were fed a HFD for 12 weeks. **a** SHP and DNMT3A occupancies at the promoter of hepatic *Fasn* determined by ChIP. **b** Methylated DNA at the *Fasn* promoter determined by MeDIP. **c** Levels of *Fasn* mRNA measured by RT-qPCR. **d** FASN and SREBP1 protein levels in whole cell liver extracts determined by IB. Results from two mice in each group are shown and the IB analysis was done once. **a–c** The mean values ± SD are plotted ($n = 5$ mice). Statistical significance was determined by (**a**, **b**) two-way ANOVA with the Tukey post-test or (**c**) two-tailed Student's $t$ test. **$P < 0.01$.

results in excessive lipogenesis, which contributes to NAFLD[3]. In this study, we show that lipogenic gene expression in the late fed state does not decrease passively because of decreased insulin/glucose-mediated stimuli, but is actively repressed physiologically by FGF15/19. Mechanistically, SHP, activated by FGF15/19 signaling-mediated phosphorylation, recruits DNMT3A to lipogenic genes that have been activated by SREBP1, resulting in DNA methylation and epigenetic repression of the genes. In contrast, in NAFLD patients and obese mice, binding of SHP and DNMT3A at the lipogenic genes and DNA methylation is decreased, suggesting that a dysregulated FGF15/19-SHP-DNMT3A axis contributes to the elevated lipogenesis that has been observed in obesity[3,4] (Model, Fig. 9e).

Hepatic lipogenesis is epigenetically regulated by histone/chromatin modifying proteins in response to nutritional and hormonal cues. Sul and colleagues showed that *Fasn* gene transcription is activated upon feeding/insulin signaling by coordinated actions of key regulatory proteins, such as USF-1 and BAF60c[1,25,26]. SETDB2 lysine methyltransferase was shown to link fasting-induced glucocorticoid signaling to inhibition of SREBP1 activation of lipogenesis[27] and the lipogenic function of SREBP1 is also inhibited by fasting-sensing SIRT1[45]. Epigenetic regulation of lipogenic genes by histone modifications is, thus, a key mechanism for the maintenance of lipid homeostasis, but our findings show that DNA methylation mediated by an FGF19-SHP-DNMT3A axis is also an important contributor to epigenetic repression of lipogenesis. Hepatic lipogenesis is also regulated in response to circadian cues. Lazar and colleagues showed that global recruitment of HDAC3 to lipogenic genes by the circadian nuclear receptor, Rev-erbα[46], displays a diurnal rhythm[28]. Notably, plasma FGF15/19 levels fluctuate in a circadian manner[13,14] and SHP also plays an important role in circadian regulation of TG metabolism by the Clock gene[47]. A recent study showed DNA methylation by DNMT1 and DNMT3A reciprocally regulate the circadian period[48]. It is, thus, highly possible that epigenetic repression of lipogenesis by the FGF15/19-SHP-DNMT3A axis displays a diurnal rhythm.

FGF15/19 has received great attention as a therapeutic target for NAFLD and diabetes, as well as, cholestatic liver disease[7–9]. Despite intensive research efforts, the molecular mechanisms by which FGF15/19 lowers lipid levels are not clearly understood, although FGF19 was shown to inhibit insulin-mediated induction of *Fasn* in rat hepatocytes[49]. In the present study, recruitment of DNMT3A by SHP at lipogenic genes, DNA methylation at the genes, and their repression in the late fed state were largely

abolished in FGF15-KO mice, revealing FGF15/19 as a physiological repressor of lipogenesis. While this study focuses on FGF15/19, other postprandial hormones or regulatory pathways may contribute to late fed-state repression of lipogenic genes. However, feeding-mediated increases in DNA methylation at the *Fasn* promoter and gene repression were severely blunted in FGF15-KO mice, suggesting that FGF15/19 likely plays a major role in the physiological repression of lipogenesis. FGF15/19-induced cellular pathways, other than through the SHP-DNMT3A axis, may also have a role in the repression. Since FGF15/19 enhances FXR functions via phosphorylation[36,37] and FXR induces expression of *Fgf15/19 and Shp*[7,8], FXR could also have a role in the FGF19-mediated repression of lipogenesis. Further, FGF19-activated SHP reduces hepatic TG levels by regulating phosphatidylcholine levels[21]. The blunting of FGF19 effects on lipogenic gene expression and lipogenesis in SHP-KO mice and in DNMT3A-downregulated mice support the conclusion that the SHP-DNMT3A pathway is likely a major contributor to the repression of lipogenesis by FGF15/19, but do not exclude a role for other FGF15/19-induced cellular pathways.

Global DNA methylome and transcriptome studies from NAFLD patients have shown that DNA methylation is reduced at lipid metabolic genes, such as *FASN, ACLY and GPAT1*[31–34]. However, the mechanisms underlying the low DNA methylation at lipogenic genes in obesity are poorly understood. In the present study, occupancy of SHP and DNMT3A and DNA methylation at the lipogenic genes were substantially reduced in NAFLD patients, although hepatic protein levels of DNMT3A and SHP expression are unchanged. FGF15/19-mediated activation of SHP via phosphorylation is important for its nuclear localization[21,38] and FGF15/19 signaling was shown to be impaired in NAFLD patients and obese mice[21,42,43], which is consistent with reduced phosphorylation of SHP and consequently, decreased nuclear levels of SHP in the patients and obese mice. We further show that this FGF15/19-induced phosphorylation is important for the interaction of SHP with DNMT3A. Our findings suggest that DNA hypomethylation at lipogenic genes in NAFLD patients may, thus, be in part due to decreased DNMT3A occupancy at lipogenic genes as a result of a defective FGF15/19-SHP axis, which contributes to increased gene expression.

In the present study, viral-mediated overexpression of SHP in obese mice substantially reduces liver TG levels by inhibiting SREBP1 and lipogenesis, which differs with the results from previous transgenic SHP mouse studies. Transgenic mice constitutively overexpressing SHP (6-7 fold) exhibit obesity and

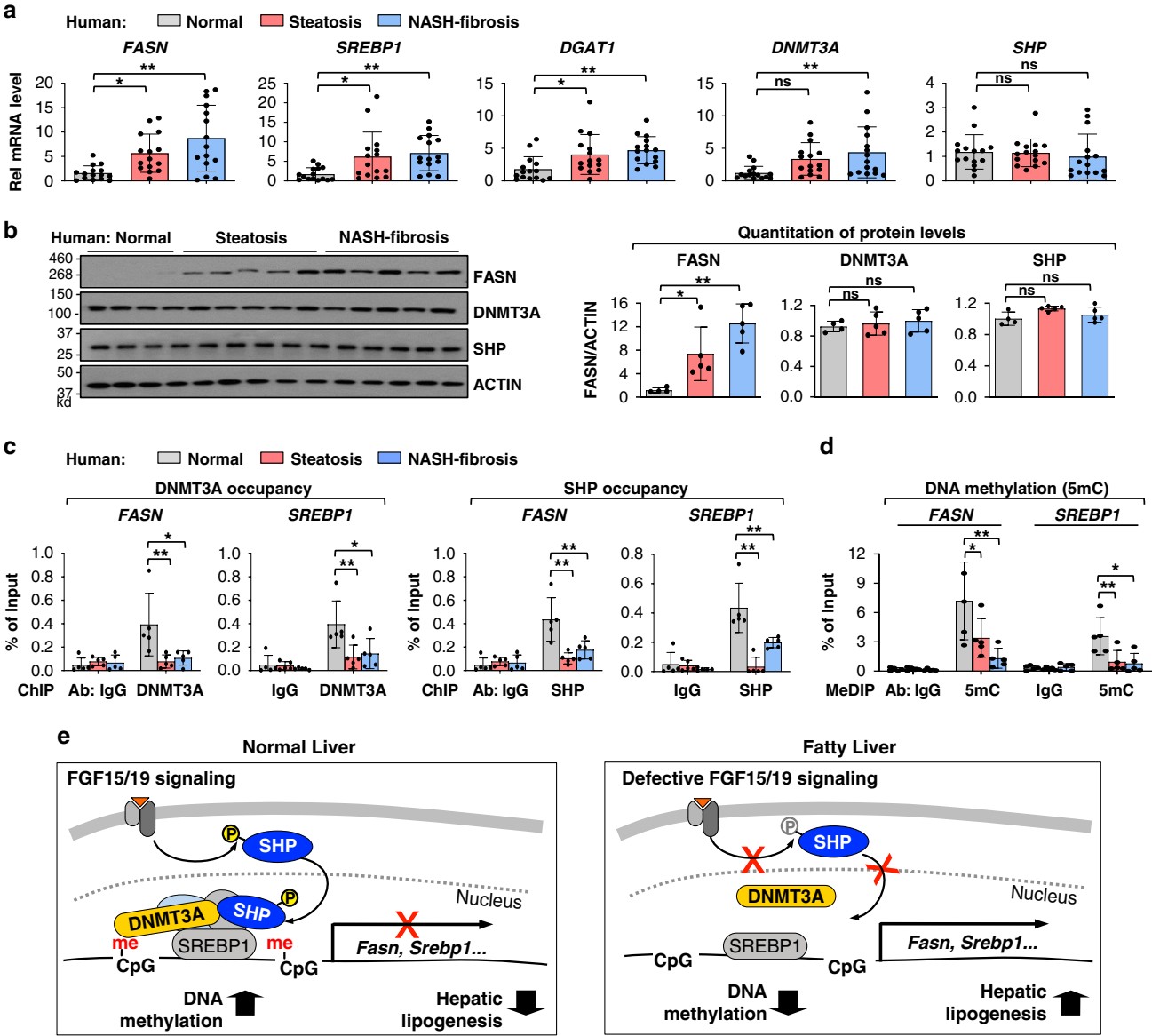

**Fig. 9 In NAFLD patients, occupancy of DNMT3A and SHP and DNA methylation at lipogenic genes are decreased, which is consistent with increased gene expression. a** Hepatic mRNA levels of the lipogenic genes, *DNMT3A* and *SHP*, determined by RT-qPCR for 15 normal individuals and 15 simple steatosis and 15 severe NASH-fibrosis patients. **b** Protein levels of FASN, DNMT3A, and SHP in liver extracts of normal subjects and NAFLD patients. Liver extracts from 15 individuals were randomly combined to form 4 (normal, 3 or 4 individuals) or 5 (steatosis and NASH, 3 individuals) pooled samples. **c** Liver extracts from 15 individuals were randomly combined to form 5 pooled samples, each containing 3 individual samples. DNMT3A and SHP occupancies at the *FASN* and *SREBP1* promoters. **d** Methylated DNA at the *FASN* and *SREBP1* promoters. **a–d** The mean and standard deviation are plotted. **a** ($n = 15$ individuals), **b** ($n = 4$, normal, $n = 5$, patients), **c**, **d** ($n = 5$, individuals). Statistical significance was determined by (**a**, **b**) one-way ANOVA or (**c**, **d**) two-way ANOVA with the Tukey post-test. *$P < 0.05$, **$P < 0.01$, ns, statistically not significant. **e** Model: An FGF15/19-SHP-DNMT3A axis represses hepatic lipogenesis physiologically but may be defective in fatty liver. Normal liver (left): FGF15/19 signaling-activated SHP via phosphorylation recruits DNMT3A to SREBP1-bound lipogenic genes, which results in DNA methylation and epigenetic repression of lipogenic genes in the late fed state. Fatty liver (right): In NAFLD patients and obese mice, impaired FGF15/19 signaling leads to decreased phosphorylation of SHP, resulting in decreased nuclear localization and occupancy of SHP at lipogenic genes, and consequently, decreased recruitment of DNMT3A and DNA methylation, which is associated with increased gene expression.

hepatic steatosis[50]. SHP-KO mice are lean but display insulin resistance[51], and *Shp* deletion in the liver was also shown to prevent hepatic steatosis in mice fed a western diet[52]. In contrast, bile acid-activated FXR reduces hepatic TG levels by inhibiting SREBP1 via SHP[53] and that viral-mediated overexpression of SHP in obese mice decreases hepatic TG levels in part by regulating phosphatidylcholine levels[21] and by inhibiting lipogenesis as observed in the current study. The reason for these opposite effects of germline SHP-KO compared to downregulation of SHP

in adult mice on energy metabolism is unclear and warrants further investigation. It might be explained by possible compensatory changes in response to constant overexpression or lack of SHP during development that alter energy metabolism in the transgenic mice.

Our studies show that liver-specific downregulation of DNMT3A in mice partially reverses FGF19-mediated DNA methylation at the *Fasn* promoter and repression of lipogenesis. Further, in obese mice, decreased lipogenic gene expression, de

novo lipogenesis, and decreased hepatic steatosis and glucose tolerance mediated by SHP were also dependent on hepatic DNMT3A. In contrast to our findings of beneficial effects of hepatic DNMT3A, a recent study has shown that adipocyte DNMT3A mediates insulin resistance in adipose tissue[29]. Further, adipo-specific DNMT3A-KO mice were protected from diet-induced insulin resistance and glucose intolerance, adipocyte *Fgf21* was identified as a target gene inhibited by DNMT3A, and intriguingly, DNA methylation at the *Fgf21* gene was elevated in adipose tissue of diabetic patients[29]. Thus, tissue-specific effects of DNMT3A function, which are beneficial in liver but detrimental in adipose tissue, may result in undesired effects if DNMT3A is targeted systemically to treat obesity-associated diseases. Tissue-specific targeting to activate or inhibit DNMT3A in liver or adipocytes, respectively, or targeting the FGF19-SHP axis may be required for a beneficial response.

In summary, we demonstrate that FGF15/19 and FGF15/19-activated SHP actively repress hepatic lipogenesis in the late fed state by recruiting DNMT3A to lipogenic genes. This active repression both limits the increase in mRNA and protein levels of lipogenic genes observed after feeding and begins the transition from the fed- to a fasted state. However, this FGF15/19-SHP-DNMT3A axis is likely dysregulated in NAFLD patients and obese mice, which contributes to decreased DNA methylation at the lipogenic genes and increased gene expression[31–34]. NAFLD is the most common chronic liver disease that can progress to fatal cirrhosis and cancer, but etiology of disease development is poorly understood and therapeutic options are limited[3]. The FGF15/19-SHP-DNMT3A axis identified in this study, thus, may provide therapeutic options for NAFLD and other obesity-associated disease.

## Methods

**Materials and reagents**. Antibodies for SHP (sc-30169), DNMT3A (sc-10232), DNMT3B (sc-10236), DNMT1 (sc-10222), SREBP1 (sc-8984), tubulin, and lamin (sc-20680) were purchased from Santa Cruz Biotechnology; for FASN (#3180) and actin (#4970) from Cell Signaling; and for 5-methylcytosine (5mC, A3001-200) from Zymo Research. Information on antibody dilution is provided in Supplementary Table 1. Deuterium oxide (113366) was purchased from Millipore Sigma.

**Animal experiments**. Male mice were housed at 23 °C, 12 h/12 h light/dark cycle, and 50% humidity, and food and water were available ad libitum. SHP-KO[54] and FGF15-KO[17] mice have been described. For fasting/refeeding experiments, 8 week-old male SHP-KO mice and control C57BL/6 mice, or FGF15-KO mice and control littermates were fasted overnight and then fed normal chow for the times indicated. For adenoviral and AAV injections, 0.5 to 1.0 × 10⁹ active viral particles and 1–2 × 10¹¹ genome copies/body weight, respectively, were injected via the tail vein. SHP-WT or SHP-T55A were expressed in mice by injection via the tail vein with Ad-SHP-WT or Ad-SHP-T55A, respectively. One week later, the mice were fasted for 4 h and sacrificed. Adenovirus expressing SHP, Ad-SHP-WT or SHP-T55A, have been described[18,21,39]. For liver-specific downregulation of DNMT3A, C57BL/6 mice were injected with AAV-control or AAV-shRNA for DNMT3A via the tail vein and 6 weeks later were fasted overnight and then treated with 1 mg/kg FGF19 for 2 h for DNA methylation and pre-mRNA studies, for 6 h for mRNA studies, and for 2 days for lipogenesis studies. To induce NAFLD, C57BL/6 mice were fed a high-fat (60% fat; Research Diets) diet with 25% fructose in the drinking water for 8 weeks. Obese mice were injected with viruses and 4 weeks later were sacrificed after 4 h of fasting. For GTT, mice were fasted overnight and injected i.p. with 2 g/kg glucose, and blood glucose levels were measured using an Accu-Chek Aviva glucometer (Roche). Liver or serum TG levels were measured using a commercial kit (Abcam, ab65336). Liver tissue was frozen in OCT compound, sectioned, and stained with H&E and Oil Red O.

**Study approval**. Animal protocols were approved by the Institutional Animal Care and Use Committee and experiments were approved by the Institutional Biosafety Committee of the University of Illinois at Urbana-Champaign. All experiments were performed in accord with the ethical guidelines for animals of the National Institutes of Health (NIH).

**RNA-seq**. C57BL/6 mice were fasted overnight and treated with 1 mg/kg FGF19 or vehicle for 6 h. The RNAs from livers of fasted mice ($n = 3$/group) were prepared using the RNeasy mini prep kit (Qiagen). The cDNA libraries were sequenced

using an Illumina HiSeq2000 (Illumina, San Diego, CA) to produce paired-end 100 bp reads. One library of reads per biological sample was examined for sequencing errors prior to mapping. Sequencing alignment was performed by STAR ver 2.6.0c, and the sequence data were analyzed with the edgeR-based R, version 3.5.0 (R Core Development Team, Vienna, Austria) pipeline. Differences between the control and FGF19-treatment samples with a $p < 0.01$ were considered significant. Gene ontology (G/O) was analyzed by the DAVID 6.8. For validation, mRNA levels of selected genes were determined by RT-qPCR and normalized to 36B4 levels with the primers listed in Supplementary Table 2.

**Mouse hepatic CpG islands and SHP cistrome**. The positions of CpG islands in the mouse genome were downloaded from (http://hgdownload.soe.ucsc.edu/goldenPath/mm9/database/cpgIslandExt.txt.gz) and the SHP cistrome was described in previous liver ChIP-seq studies[18]. SHP binding peaks that overlapped with CpG islands for at least 1 bp were considered overlapping.

**Bisulfite sequencing**. Mouse genomic DNA was subjected to DNA denaturation and bisulfite conversion processes using the EZ DNA Methylation-Direct Kit (Zymo Research, D5030). The bisulfite-modified DNA was amplified by PCR using the primers designed using EpiDesigner 1.0 (http://www.epidesigner.com/start3.html). The amplified DNA was sequenced and the sequence data were analyzed by BISMA 1.0 software (http://services.ibc.uni-stuttgart.de/BDPC/BISMA/) using default threshold settings.

**Hepatic de novo lipogenesis (DNL)**. Mice were injected i.p. with 27 ul/g of $D_2O$ and then were given drinking water that contained 4% $D_2O$. After 2 days, mice were sacrificed and lipids were extracted from liver tissue by Folch's extraction procedure. Briefly, lipids were extracted by homogenization in chloroform/methanol/water (8:4:3) followed by centrifugation. The extracted lipids in the organic phase were analyzed by mass spectroscopy to determine composition of fatty acids in the lipid sample. The percentage of fatty acids formed by DNL during $D_2O$ exposure was calculated.

**Cell culture**. Primary mouse hepatocytes (PMHs) were isolated by collagenase (0.8 mg/ml, Sigma, Inc) perfusion through the portal vein of mice anesthetized with isoflurane. Hepatocytes were filtered through a cell strainer (100 μm nylon, BD), washed by centrifugation with M199 medium, resuspended in M199 medium, centrifuged through 45% Percoll (Sigma, Inc.), and cultured in M199 medium containing 10% FBS.

**ChIP, MeDIP, and CoIP**. For ChIP assays, liver tissue was minced, washed twice in PBS, and then incubated with 1% formaldehyde for 10 min at 37 °C. Glycine was added to 125 mM for 5 min at room temperature. The tissue was washed in PBS, resuspended in ice-cold IP lysis buffer (10 mM KOH-Hepes, pH 7.9, 1.5 mM MgCl₂, 10 mM KCl, 0.2% Nonidet P40, 1 mM EDTA, 5% sucrose) and homogenized with a Dounce homogenizer. The homogenate was layered over a sucrose cushion (10 mM Tris-HCl pH 7.5, 15 mM NaCl, 60 mM KCl, 1 mM EDTA, 10% sucrose) and centrifuged at 3000 × g for 5 min. The nuclear pellet was resuspended in sonication buffer (50 mM Tris-HCl, pH 8.0, 2 mM EDTA, and 1% SDS). For MeDIP, genomic DNA was isolated from the liver using a DNA isolation kit (Qiagen). The resuspended nuclear pellet or genomic DNA in sonication buffer with protease inhibitors was sonicated using a QSonica 800R2-110 at amplitude setting 70% with sonication pulse rate 15 s on and 45 s off. The chromatin sample (500 μl) was precleared by incubation with Protein G–Sepharose slurry for 1 h and immunoprecipitated using 1–2 μg of antibody or IgG overnight at 4 °C. The immune complexes were collected by incubation with a Protein G Sepharose slurry for 1 h, washing with 0.1% SDS, 1% Triton X-100, 2 mM EDTA, 20 mM Tris-HCl, pH 8.0, three times containing successively 150 mM NaCl, 500 mM NaCl, or 0.25 M LiCl, and then eluted. The chromatin samples were incubated overnight at 65 °C to reverse the crosslinking. DNA was isolated and enrichment of gene sequences in the immunoprecipitates was determined by RT-qPCR. For re-ChIP, chromatin samples were immunoprecipitated with the first antibody, the beads were washed, chromatin was eluted with 10 mM DTT, diluted 20X with 20 mM Tris-HCl, pH 8.0, 150 mM NaCl, 2 mM EDTA, 1% Triton X-100, and immunoprecipitated with the second antibody. Sequences of primers used for the qPCR are in Supplementary Table 2. For CoIP, cell extracts were prepared by brief sonication in CoIP buffer (50 mM Tris–HCl, pH 8.0, 150 mM NaCl, 2 mM EDTA, 0.5% NP-40, 5% glycerol). The samples were incubated with 1–2 μg of antibodies for 3 h and 30 μl of a 25% protein G agarose slurry was added. One hour later, beads were washed with CoIP buffer three times and bound proteins were detected by IB.

**Luciferase reporter assay**. Hepatocytes were transfected with expression plasmids as indicated, including those for DNMT3A-WT and a catalytically inactive mutant DNMT3A-C706S[29], for 72 h and the cells were incubated in serum-free medium overnight and treated with or without FGF19 (50 ng/ml) for 6 h. The values for luciferase activities were normalized to β-galactosidase activities.

**RT-qPCR**. Total RNA was isolated using Trizol, cDNA was synthesized and RT-qPCR was performed with the iCycler iQ (Bio-Rad). The amount of mRNA or pre-mRNA was normalized to that of 36B4 mRNA. To measure pre-mRNA levels of lipogenic genes, forward primers in exon 1 and reverse primers in intron 1 were used for the qPCR. Sequences of primers are in Supplementary Table 2.

**NAFLD patient study**. Liver specimens from 15 unidentifiable normal individuals without liver disease or steatosis or severe NASH-fibrosis patients were obtained from the Liver Tissue Cell Procurement and Distribution System (LTCDS) that operates under a contract from the NIH. The LTCDS obtains organ transplant tissues from regional centers in the United States that have obtained the human subjects' approval to provide portions of the resected pathologic liver for which the transplant is performed or portions from normal livers rejected for transplant. Because the specimens or data were not collected specifically for this study and no one on our study team has access to the subject identifiers linked to the specimens or data, this study is not considered human subjects research and ethical approval was not required (See section 46.104 in Part 46—Protection Of Human Subjects in the Electronic Code of Federal Regulations at the following link: https://www.ecfr.gov/cgi-bin/retrieveECFR?gp=&SID=3cd09e1c0f5c6937cd9d7513160fc3f&pitd=20180719&n=pt45.1.46&r=-PART&ty=HTML#se45.1.46_1104). Protein levels ($n = 4$–5, each pooled from 3 to 4 individual liver samples/lane) were detected by IB, mRNA levels ($n = 15$) by RT-qPCR, and protein occupancy and DNA methylation by ChIP and MeDIP, respectively.

**Statistical analysis**. GraphPad Prism 8 (version 8.2) was used for data analysis. Statistical significance was determined by the Student's two-tailed t-test or one- or two-way ANOVA with the Tukey post-test for single or multiple comparisons as appropriate.

**Reporting summary**. Additional information on research design is available in the Nature Research Reporting Summary linked to this article.

## Data availability
The RNA-seq data in the present study and the published SHP ChIP-seq data[18] are deposited in the Gene Expression Omnibus (GEO) database with the Accession Numbers GSE158359 and GSE74913, respectively. Any other data are available upon reasonable request from the corresponding author. Source data are provided with this paper.

## Code availability
There is no custom computer code or algorithm used to generate the results in this paper.

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

## Acknowledgements

The authors thank H. Eric Xu at Van Andel Research Institute for purified recombinant FGF19 and Sona Kang at University of California at Berkeley for the DNMT3A (C706S) plasmid. We also thank the Liver Tissue Cell Distribution System, University of Minnesota (NIH Contract # HHSN276201200017C) for liver specimens of NAFLD patients. Petri dish, mouse, and liver images in the figures are unmodified images provided by Servier Medical Art by Servier, licensed under a Creative Commons Attribution 3.0 Unported License [https://creativecommons.org/licenses/by/3.0/legalcode] This study was supported by an American Heart Association Scientist Development Award (16SDG27570006) to YK and by grants from the National Institutes of Health (DK062777 and DK095842) to JKK.

## Author contributions

Y.K., S.S., and J.K. designed research; Y.K, S.S., and Bo.K. performed experiments; Y.K, S.S., Bo.K., By.K., G.G., and J.K. analyzed data; Y.K., Y.Z., and J.M. performed genomic analyses; and Y.K, By.K., and J.K. wrote the paper.

## Competing interests

The authors declare no competing interests.
