## [Peer Review File · Nature Communications]

Reviewers' comments:

Reviewer #2 (Remarks to the Author):

Kim et al. describe an interesting cooperative repression of lipogenic genes via SHP dependent recruitment of the DNA methyltransferase Dnmt3a to lipogenic genes. In contrast to the better characterized interaction of SHP with nuclear receptors, SHP interacts with SREBP1 at these lipogenic targets. Overall the results provide solid support for these conclusions. However, there is an important issue that remains to be addressed before this story is complete. Although DNA methylation is now recognized to be reversible, in most cases it is considered a relatively stable chromatin mark. For the acute increase in methylation in the late-fed state described here to be physiologically relevant, the basal level of methylation must be reset prior to the SHP/Dnmt3a recruitment. Thus, to complete the regulatory circuit is essential to not only show the late-fed increase in the 5mC mark, but also document its subsequent erasure in the fasted state. This is touched on in Fig. 3e, but it should include analysis of 5mC levels in the FASN and Cyp7A1 promoters, and also a negative control stably methylated promoter, over a full fasting-feeding-fasting cycle to document the presumed demethylation necessary to ready the lipogenic promoters to respond appropriately again to the feeding stimulus.

There are also 2 important issues from the discussion. The first is that the Lazar lab showed that hepatic lipogenesis is strongly responsive to a circadian Rev-ErbA dependent epigenetic cycle. At least some discussion of the potential relationship of the current results with that important prior study should be considered in the discussion. The second is that all of the results and discussion predict that the SHP knockouts should have fatty liver due to dysregulated lipogenesis, but that is not the consensus of the field. This issue deserves comment, and the perspective on the overall results could be improved.

Reviewer #3 (Remarks to the Author):

Kim et al. report interesting and timely data showing the connection of growth factors, transcription factors and epigenetic effects (DNA methylation) in the regulation of liver lipogenesis genes in a late fed-state. Many data points are strong and convincing and the overall story is very appealing. Most experiments were well-designed and conducted. I think an appropriately revised version of this manuscript could become a very strong candidate for publication in Nat. Comm.

Major comments

- 1) p. 5: Specify how pre-mRNA was identified.
- 2) The source of the SHP KO mice is not specified. If they were made here, details must be provided and the KO validated. Similarly, the FGF15 KO mice are insufficiently described and not validated in the given reference.
- 3) Fig. 2f: A large fraction of FGF19 target genes are not overlapping with SHP. This goes against the hypothesis that SHP is the major effectors of FGF19.
- 4) Key DNA methylation data must be validated by Bisulfite sequencing of exemplary regions.
- 5) The methylation increase in Fig. 3e, Fasn, littermate is not convincing. Additional data must be provided.
- 6) The effect reported in Fig. 4g is not convincing. Additional data must be provided.
- 7) p. 17: It is inappropriate to present data first time in the discussion. Suppl. Fig. 7 must be presented and described already in the results part.

Minor comments

- 8) p. 3: It is stated the overall actions of FGF19 resemble insulin, but in the description, it rather sounds as if FGF19 help to terminate insulin action.
- 9) p. 4: DNMT1 does not preferentially bind hemimethylated DNA, but it preferentially methylates it.
- 10) p. 4: Remove “reversible” in “catalyse reversible DNA”. DNA methylation is reversible but DNMT3 enzymes introduce it.
- 11) Suppl. Fig. S1 reports important data for the story of the manuscript. It would be better presented in the main manuscript, e.g. integrated into Fig. 1.
- 12) Fig. 2A Specify the basis of the log.
- 13) p. 7 end of first paragraph: Correlation with DNA methylation cannot be inferred at this point.
- 14) p. 7: “islands of CpG” correct to “CpG islands”
- 15) Fig. 2e: I assume the location of CpG islands is plotted in blue, not DNA methylation data. If so, remove 5mC from the second line in the heading. Replace the red “peak” by a red line. Change text to “SHP ChIP-seq peaks” and remove black text.
- 16) Provide explanation for light and dark grey bars in Fig. 4d-g.
- 17) Provide explanation for “ND” samples in Fig. 7.

18) Suppl. fig. 6 should be presented in the main manuscript.

Responses to the reviewers

Reviewer #2 (Remarks to the Author)

Kim et al. describe an interesting cooperative repression of lipogenic genes via SHP dependent recruitment of the DNA methyltransferase Dnmt3a to lipogenic genes. In contrast to the better characterized interaction of SHP with nuclear receptors, SHP interacts with SREBP1 at these lipogenic targets. Overall the results provide solid support for these conclusions. However, there is an important issue that remains to be addressed before this story is complete.

Although DNA methylation is now recognized to be reversible, in most cases it is considered a relatively stable chromatin mark. For the acute increase in methylation in the late-fed state described here to be physiologically relevant, the basal level of methylation must be reset prior to the SHP/Dnmt3a recruitment. Thus, to complete the regulatory circuit is essential to not only show the late-fed increase in the 5mC mark, but also document its subsequent erasure in the fasted state. This is touched on in Fig. 3e, but it should include analysis of 5mC levels in the *Fasn* and *Cyp7A1* promoters, and also a negative control stably methylated promoter, over a full fasting-feeding-fasting cycle to document the presumed demethylation necessary to ready the lipogenic promoters to respond appropriately again to the feeding stimulus.

Response: Thanks for this important comment. As requested by the reviewer, we have analyzed the changes in 5-mC levels throughout the full fasting-feeding-fasting cycle for *Fasn* and *Cyp7a1* and additionally for *Esr1*, a highly methylated gene in the liver (Maegawa et al., Genome Research, 2010). The levels of 5-mC at *Fasn* increase after refeeding and return to the initial fasting levels after the second fasting. In contrast, DNA methylation at *Cyp7a1* was not detected and no changes in 5-mC levels at *Esr1* were detected throughout the cycle. The new data are now presented in Fig. 3f.

There are also 2 important issues from the discussion. The first is that the Lazar lab showed that hepatic lipogenesis is strongly responsive to a circadian Rev-ErbA dependent epigenetic cycle. At least some discussion of the potential relationship of the current results with that important prior study should be considered in the discussion.

Response: We have incorporated a discussion of the circadian regulation of lipogenesis reported by the Lazar group and other published circadian studies on FGF15/19, SHP, and DNMT3A, which suggest a possible role for this axis in the circadian control of lipogenesis (page 16).

The second is that all of the results and discussion predict that the SHP knockouts should have fatty liver due to dysregulated lipogenesis, but that is not the consensus of the field. This issue deserves comment, and the perspective on the overall results could be improved.

Response: We thank the reviewer for this comment since it highlights an important difference between germ-line knockout (KO) of SHP and viral-mediated knockdown (KD) of SHP in adult mice. In studies that are part of a manuscript in preparation, viral-mediated liver-specific LKD

of SHP in adult mice and germ-line LKO of SHP have opposite effects on energy metabolism. While SHP-LKO mice displayed decreased body weight compared to control mice, SHP-LKD mice showed increased body weight without changes in food intakes compared to control mice, even under normal chow diet (See the “Information for Reviewers Only” figure). Presumably, the chronic germ-line depletion of SHP results in compensatory changes to the loss of SHP. Discussion of this issue is added in the Discussion section (page 18), noting that our results in the KD mice differ with the previous findings from germ-line SHP transgenic mouse studies.

Figure (Reviewers only). Liver-Specific SHP knock-out (LKO) vs liver-specific downregulation (LKD) mice. SHP-LKO mice were generated by breeding mice with a floxed SHP gene (SHP f/f) with Alb-Cre mice. SHP-LKD mice were generated by infecting 8 week old SHP f/f mice with AAV-TBG-Cre 12 weeks under normal chow diet. (a) Picture of a representative mouse in each group. (b) Body weight. (c) Food intake. The mean and standard deviation are plotted (n = 5). Statistical significance was determined Student's t-test. *P < 0.05, **P < 0.01.

Reviewer #3 (Remarks to the Author)

Kim et al. report interesting and timely data showing the connection of growth factors, transcription factors and epigenetic effects (DNA methylation) in the regulation of liver lipogenesis genes in a late fed-state. Many data points are strong and convincing and the overall story is very appealing. Most experiments were well-designed and conducted. I think an appropriately revised version of this manuscript could become a very strong candidate for publication in Nat. Comm.

Major Comments:

1) p. 5: Specify how pre-mRNA was identified.

Response: We have added in the Methods (RT-qPCR section) that pre-mRNA of lipogenic genes in Figure 1 was detected using a forward primer in exon 1 and a reverse primer in intron 1, so that only pre-mRNA sequences are amplified in the RT-qPCR.

2) The source of the SHP-KO mice is not specified. If they were made here, details must be provided and the KO validated. Similarly, the FGF15 KO mice are insufficiently described and not validated in the given reference.

Response: The SHP-KO mice and FGF15-KO mice have been characterized previously. We have added a sentence in the Methods (first sentence, Animal Experiments section) with references to previous studies.

3) Fig. 2f: A large fraction of FGF19 target genes are not overlapping with SHP. This goes against the hypothesis that SHP is the major effectors of FGF19.

Response: In the original manuscript, we probably over-emphasized the role of SHP in mediating the global actions of FGF19. We show that SHP plays a prominent role in regulating the key lipogenic gene, *Fasn*, but based on the large fraction of FGF19 downregulated genes that do not have SHP binding sites, many other hepatic functions of FGF19 are SHP-independent. We have, therefore, deleted the Venn diagram data presented in Figure 2 in the original manuscript and have revised our discussion to reflect the more specific role of SHP in the effects of FGF19 on hepatic lipid metabolism, particularly lipogenic genes.

4) Key DNA methylation data must be validated by Bisulfite sequencing of exemplary regions.

Response: We thank the reviewer for this constructive suggestion. In response to the reviewer's suggestion, we have conducted additional in vivo experiments in which the methylation at the *Fasn* promoter was analyzed by bisulfite sequencing in fasting and refed mice and also in mice treated with FGF19 or vehicle. We obtained results consistent with the MeDIP assays and the new data are now presented in Fig. 2h and 3e.

5) The methylation increase in Fig. 3e, *Fasn*, littermate is not convincing. Additional data must be provided.

Response: In response to the previous comment, we analyzed the *Fasn* promoter in fasting and refed mice by bisulfite sequencing, which more clearly shows the increase in methylation at specific sites in the refed mice.

6) The effect reported in Fig. 4g is not convincing. Additional data must be provided.

Response: To complement the de novo lipogenesis data, we further analyzed hepatic TG levels (new data in Fig. 4h), which shows that downregulation of DNMT3 blocks the FGF19-mediated ~50% decrease in TG levels.

7) p. 17: It is inappropriate to present data first time in the discussion. Suppl. Fig. 7 must be presented and described already in the results part.

Response: As suggested, we have moved the description of the results in Suppl. Fig. 7 from the Discussion section to the Results section (page 7).

Minor Comments:

8) p. 3: It is stated the overall actions of FGF19 resemble insulin, but in the description, it rather sounds as if FGF19 help to terminate insulin action.

Response: Thanks for this critical comment. The overall metabolic actions of FGF15/19 resemble insulin, but in most cases are independent of insulin action (Kir, et al., Science, 2011). In the present study, we found that FGF15/19 is important for termination of lipogenic effects of insulin in the late fed-state, which is consistent with the observed reduction of lipogenesis by FGF19 while in contrast insulin increases lipogenesis. The different effects on lipogenesis of insulin and FGF19, as well as the reduction of TGs by FGF19 are noted in the sentence in the Introduction (page 3).

9) p. 4: DNMT1 does not preferentially bind hemimethylated DNA, but it preferentially methylates it.

Response: We have corrected this error.

10) p. 4: Remove “reversible” in “catalyze reversible DNA”. DNA methylation is reversible but DNMT3 enzymes introduce it.

Response: Corrected as suggested. We have deleted “reversible”.

11) Suppl. Fig. S1 reports important data for the story of the manuscript. It would be better presented in the main manuscript, e.g. integrated into Fig. 1.

Response: As suggested by the reviewer, we have moved Fig. S1 into the main manuscript (Fig. 1d,f).

12) Fig. 2A Specify the basis of the log.

Response: We have modified the legend for the X-axis to “Log₁₀FC”.

13) p. 7 end of first paragraph: Correlation with DNA methylation cannot be inferred at this point.

Response: We have altered this conclusion and have removed the correlation reference.

14) p. 7: “islands of CpG” correct to “CpG islands”

Response: Corrected as suggested.

15) Fig. 2e: I assume the location of CpG islands in plotted in blue, not DNA methylation data. If so, remove 5mC from the second line in the heading. Replace the red “peak” by a red line. Change text to

“SHP ChIP-seq peaks” and remove black text.

Response: We have modified Fig. 2e as suggested by the reviewer.

16) Provide explanation for light and dark grey bars in Fig. 4d-g.

Response: We have changed the color scheme to white and blue bars, which are defined below Fig. 4a in the figure.

17) Provide explanation for “ND” samples in Fig. 7.

Response: We have defined ND as normal chow diet in the Figure 7 legend.

18) Suppl. fig. 6 should be presented in the main manuscript.

Response: As suggested, we have presented our model in Figure 9e.

We thank both reviewers for their helpful constructive comments. We believe we have addressed all of the concerns and issues raised by the reviewers and our revisions have greatly strengthened the manuscript. We hope that our revised manuscript will now be acceptable for publication.

REVIEWERS' COMMENTS

Reviewer #2 (Remarks to the Author):

This revised manuscript adequately addresses prior concerns.

Reviewer #3 (Remarks to the Author):

The authors have shown that FGF15/19 regulates hepatic lipogenesis by recruitment of DNMT3A to lipogenic target genes followed by increased DNA methylation and gene silencing. This paper provides a clear and very well documented example of physiological regulation via the recruitment of a DNMT and increase in DNA methylation. It nicely illustrates that DNA methylation has important roles in the transient regulation of cellular phenotypes. This is a very interesting and very important paper. The experiments were very well planned and conducted. The results are convincing and they shed new light on the connection of cellular signalling system with chromatin based epigenome signalling. My comments were sufficiently answered and I do recommend publication of this nice work.

NCOMMS-19-36273A

“Intestinal FGF15/19 physiologically repress hepatic lipogenesis in the late fed-state by activating SHP and DNMT3A”

Responses to the reviewers

REVIEWERS' COMMENTS

Reviewer #2 (Remarks to the Author):

This revised manuscript adequately addresses prior concerns.

Reviewer #3 (Remarks to the Author):

The authors have shown that FGF15/19 regulates hepatic lipogenesis by recruitment of DNMT3A to lipogenic target genes followed by increased DNA methylation and gene silencing. This paper provides a clear and very well documented example of physiological regulation via the recruitment of a DNMT and increase in DNA methylation. It nicely illustrates that DNA methylation has important roles in the transient regulation of cellular phenotypes. This is a very interesting and very important paper. The experiments were very well planned and conducted. The results are convincing and they shed new light on the connection of cellular signalling system with chromatin based epigenome signalling. My comments were sufficiently answered and I do recommend publication of this nice work.

Response: The reviewers had no additional concerns.